# Context-Induced Activity Monitoring for On-Demand Things-of-Interest Recommendation in an Ambient Intelligent Environment

May Altulyan [1,2], Lina Yao [1,*], Chaoran Huang [1], Xianzhi Wang [3] and Salil S. Kanhere [1]

1   School of Computer Science and Engineering, University of New South Wales, Sydney, NSW 2052, Australia; m.altulayan@psau.edu.sa (M.A.); chaoran.huang@unsw.edu.au (C.H.); salil.kanhere@unsw.edu.au (S.K.)
2   College of Computer Engineering and Science, Prince Sattam Bin Abdulaziz University, Alkharj 11942, Saudi Arabia
3   School of Computer Science, The University of Technology Sydney, Ultimo, NSW 2007, Australia; xianzhi.wang@uts.edu.au (X.W.)
*   Correspondence: lina.yao@unsw.edu.au

**Abstract:** Recommendation systems are crucial in the provision of services to the elderly with Alzheimer's disease in IoT-based smart home environments. In this work, a Reminder Care System (RCS) is presented to help Alzheimer patients live in and operate their homes safely and independently. A contextual bandit approach is utilized in the formulation of the proposed recommendation system to tackle dynamicity in human activities and to construct accurate recommendations that meet user needs without their feedback. The system was evaluated based on three public datasets using a cumulative reward as a metric. Our experimental results demonstrate the feasibility and effectiveness of the proposed Reminder Care System for real-world IoT-based smart home applications.

**Keywords:** contextual bandit; IoT; recommender system

## 1. Introduction

Alzheimer's disease (AD) has been considered as the most common cause of dementia given its significant impairment of cognitive abilities, which comes with severe implications in human day-to-day activities [1]. In 2017, over 6.08 million elderly people in the United States reportedly suffer from various classes of AD with this figure potentially set to escalate to 15.0 million by 2060 [2]. This comes with huge cost implications as the management of AD and other types of dementia reportedly cost approximately $277 billion and $290 billion in 2018 and 2019, respectively  [3,4].

Generally, AD is categorised into three main stages, mild, moderate, and late or severe stages with each of the stages presenting various symptoms. Patients suffering mild AD lose only short-term memory where they experience difficulty in remembering people's names or recent events. At the mild stage, technological aids are deployed to manage the disease. Patients suffering moderate AD may suffer intense memory loss, which could impact their abilities for coordinating and handling easy tasks due to increased poor judgments and deepened confusion.

Language problems, time consideration, and significant changes in their personality are major indicators. In the last stage, patients suffering severe AD lose their comprehension and physical abilities becoming unable to talk, swallow, walk etc. Consequently, patients at this stage require intensive care from professional caregivers and family members. While the mild and moderate stages may typically last for around 3 years, the late or severe stage could continue throughout the remainder of the patient's life [1].

The widened applications of IoT-based smart-home environments birthed the idea of a recommender system, reminder care systems, which are adapted to improve the management of patients with AD. A reminder care system considers patients who suffer

from the mild stage of AD where patients begin losing short-term memory; however, they still have the ability to use such a system [1,5]. A reminder care system is designed to exploit sensory data from various sources, such as the environmental sensors, wearable sensors, and appliance sensors for effective reminder recommendations. A feedback system is not necessarily required to improve the quality of recommendations.

The scenario presented in Figure 1 reveals the importance of a reminder care system for Alzheimer patients. In the scenario, the 77-year-old woman, Aris, lives independently and she is diagnosed with the mild AD. The deployed reminder care system can monitor and recognize Aris' activity patterns and then use this pattern to automatically provide recommendations on what Aris may need based on her current activity. For example, the system is designed to promptly recommend switching off of appliances if she forgets to do so after usage. The system is subsequently improved based on her acceptance of every recommendation without needing Aris' explicit feedback.

Here, the system considers all contexts about the user and items. For example, if Aris starts to prepare a cup of coffee at midnight, no item will be recommended by the system but instead it will remind her to go back to sleep because time is considered as a key context in conducting the recommendations. Moreover, the system has the ability to learn all new patterns and ignore old patterns. For illustration, when Aris no longer adds milk to her coffee, the system will not recommend milk during this activity. This scenario could be extended to become not only a smart home application but also for m-health applications.

Again, Aris at home can be better monitored by the hospital by utilizing the data from sensors that are installed in her house to remotely monitor the development of her condition. The hospital can also provide medical advice recommendations, such as recommending a specific time for resting, recommending an exercise to be done, or recommending certain kinds of foods. Our proposed system is suitable only for the first stage of Alzheimer as, in other stages, patients face more changes in their behaviour where they require intensive care. A number of studies have focused on reminder recommender systems aimed at providing assistance to elderly people who suffer from AD.

Oyeleke et al. [6] designed a recommendations system for monitoring the daily indoor activities of seniors with mild cognitive impairment while Ahmed et al. [7] proposed a smart biomedical assisted system to help patients with Alzheimer's. In some studies, smartphone applications were developed for the provision of care services to AD patients [8–11].

However, the dynamicity in the complexities of human activities is yet to be adequately addressed, thus, delivering low-quality recommendations. Another notable issue is the increased focus on monitoring, which gives a reminder to patients while the system has to wait for patient feedback to update itself. From the Aris scenario, consider if Aris follows the following sequence of actions when starting to prepare a cup of coffee in the early morning: first, switching the coffee machine and then bringing a cup, next, filling milk, and then adding sugar.

Supposing then she picks up a cup and forgets what to do next? The system should remind her to pick up the milk. Nonetheless, if one day, she changes this pattern by deciding not to add milk in the future, the system should also cope with that. From Aris' perspective, the system should be a caregiver, i.e., to offer help only when necessary without actively requesting feedback. Therefore, as a well-designed reminder system, the system must be capable of assessing the quality of recommendations without necessarily relying on user feedback.

In one of our previous works [12], we implemented a prototype system with great consideration of the dynamicity of human activities, which was capable of detecting complex activities. Then, in [13], we presented a Reminder Care System (RCS), which, in addition to being able to learn the dynamicity of human activities, could also remind patients about their needs correctly, without requiring their feedback. The problem was formulated using a contextual bandit approach, which considers contexts as input to recommend the next action. The RCS can support Alzheimer's patients in their first stage by constructing high quality recommendations only when the user is needed. Based on the

Aris scenario, our system can be considered as caregiver where it considers all the context of the exact time for needing recommendation.

For example, if Aris is in the kitchen for preparing a cup of coffee in the midnight, our system instead of recommending an item for this activity; it will recommend for Aris to go back to her bed because the time is too late. Considering context helps the system not only recommend a correct item but also to know the suitable period of each activity. Here, we extend this work by improving our RCS system of two main aspects, the reward function calculation and updating the system. We also run more experiments on three public datasets to validate our system. The main contributions of this paper are as follows:

- Proposition of a recommender system based on the contextual bandit approach by fusing the context information from the past and current activities to recommend the correct item.
- Formulation of a reward function for automatic updates without requiring feedback from users to improve the recommendations.
- Provision of minor and major updates to help tackle the dynamicity in human activities while improving the quality of recommendations.
- Evaluation of the developed model using three public datasets.

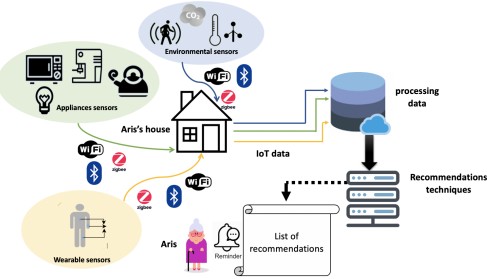

**Figure 1.** Motivation scenario.

## 2. Related work

Multiple studies related to recommender systems for the IoT are found in the literature. Some of these works exploit the traditional recommender system approaches: collaborative filtering, content-based and hybrid-based. The collaborative filtering approach makes recommendations on items for a particular user based on the ratings of previous user [14]. The authors in [15] proposed a unified collaborative filtering model based on a probabilistic matrix factorization recommender system. It utilized three kinds of relations to extract the latent factors among these relations to construct accurate recommendations.

Although the collaborative filtering approach has been adapted in numerous studies [16–19], there are potential shortcomings that make it inefficient for RSIoT particularly, in terms of large amount of data, cold start problem, and data sparsity. In content-based, instead of relying on ratings, it recommends items that are similar to the items previously targeted by the user [20].

The authors in [21] adopt a content-based solution for the recommender engine in their AGILE project, which aims to improve the health conditions of users. The CB provides a number of features compared with CF, such as creating a profile for each user that depends on the history of his rating, deciding recommended items based on the extracted features of each item, and dealing with cold start problem; nonetheless, it has a limitation where it builds its recommendations based on items and their features only without considering any additional features that can tackle the RSIoT issues.

The hybrid-based approach combines two or more approaches to build an RS, such as combining the collaborative and content-based methods where the limitations of each can be addressed [22]. The authors in [23,24] built their recommender system engine using a hybrid recommendation algorithm. Combining two approaches, such as content-based with collaborative, may address their limitations but fails to tackle RSIoT problems: the dynamicity in human activity patterns, and updating the system automatically without needing feedback from users.

Consequently, other studies started to shift for artificial intelligence techniques that are able to present data in a good way and to deal with complex interaction pattern. For example, machine learning was adopted to create a home automation framework [25]. It can be used as recommender system to assist elderly people so, they can live independently and safely. Here, the home appliances, like the TV, lights, etc. will be controlled using the user voice. In addition, If someone needs to monitor the elderly people, they can remotely control the smart devices with minimal cost and small effort.

Deep learning algorithms also have been adapted for building a recommender system platform [26]. It helps doctors to determine the rehabilitation nutrition plan for cancer patients. However, machine learning and deep learning algorithms learn from offline data, which makes a recommender system for the IoT neither able to tackle any change of the human pattern nor be updated automatically.

The reinforcement learning (RL) approach has been adapted in the implementation of recommender systems for IoT environments. RL makes it possible for the learning of the dynamics of a given environment and offers an architecture, which maximizes the long-term reward particularly for continuous record updates. In the work presented by Massimo et al. [27,28], an inverse RL was adopted to model user behaviours while Oyeleke et al. [6] designed a recommender system for monitoring the daily indoor activities for people with mild cognitive impairment using RL.

Most RL algorithms that deal with the modelling of dynamic environmental factors usually focus on matching each state for an action using different sequence of policy implementation. By considering future rewards, current actions are observed on how they influence next action. However, a notable shortcoming of RL algorithms is their inability to handle a system requiring learning and selection of the best action from different scenarios where each state is treated independently.

As we mentioned in the Aris scenario, the system works as caregiver, which means acting based on the patient behaviour by paying attention to her needs. Consequently, the RSIoT system needs to consider all context from both user and item at the same time to decide which item should be conducted at this moment. Unlike previous approaches, which were mentioned above, contextual bandit (CB) approaches can exploit both offline data and environment interactions that help in constructing recommendations with high quality.

In CB, there are three main concepts in CB: State, which defines which activity is performed by the user; Action, which represents the item that the user needs for this state; and Reward, which the system receives based on the quality of the recommended item. Contextual bandit utilizes the common features of RL by using policy to decide an action based on the context of each state. This is similar to multi-armed bandit (MAB), which focuses on the immediate reward.

As shown in Figure 2, both action and state affect the reward, which have a positive impact to increase the quality of the recommendation. In contrast, the action in RL effects not only the reward but also the action, which means that RL cannot deal with different scenarios as we mentioned before [29]. Some studies have adapted CB for their recommender systems.

Li et al. [30] utilized the contextual bandit for the recommendations of news articles. The presented algorithm, LinUCB, was reportedly applied to sparse and large data combined with other algorithms, such as e-greedy. The study presented in [31], an online learning recommender system, was developed by adapting CB where information for history students were learnt, and current students were used as context to conduct the learning recommendations to the students.

The author in [29] adapted CB to decide on action to be carried out by a robot deployed to help dementia patients with their behavioural disturbances. Zhang et al. [32] proposed a novel CB method named SAOR for online recommendations. The study offered sparse interactions, which distinguished between a negative response and non-response to improve recommendation quality. In our study, CB is adapted by utilizing three kinds of

features, context past activities, current activity, and items. A reward model for automatic update of the system requiring no feedback is also presented in this work.

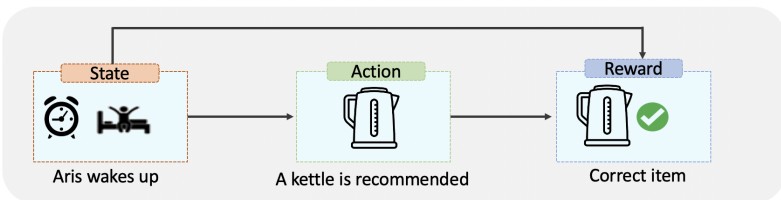

**Figure 2.** The three main concepts in the Contextual Bandit Approach are (1) State, which represents the current situation for the user; (2) Action, which provides the required item that meets the user's need; and (3) Reward, which considers feedback for the system to improve the quality of recommendations.

## 3. Contextual-Bandit-Based Reminder Care System

The system proposed in this reminder recommendation has three major stages as shown in Figure 3 and these are: complex activity recognition stage, prompt detection stage and the recommendation stage. The complex activity recognition stage hinges on three data sources, data from the wearable sensors, environmental sensory data, and home appliance usage data. The prompt detection stage utilizes the data mining approach to determine if an ongoing activity requires an item recommendation, while, at the reminder recommendation stage, the CB approach is applied to extract context from the two previous stages to recommend items to the user during an activity. The stages are further presented in the subsections below.

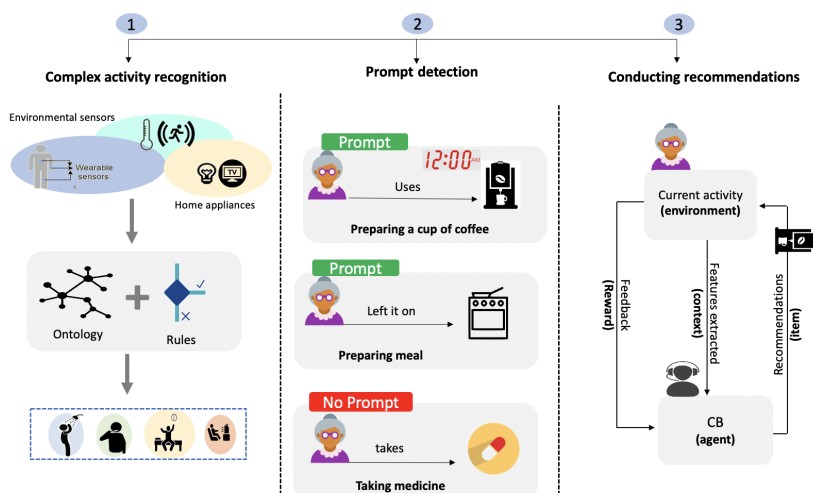

**Figure 3.** Overview of the proposed methodology.

### 3.1. Complex Activity Detection

For a system to recommend appropriately, it needs to be aware of the user's activities to enable it effectively carry out desired recommendations. Despite the extensive studies on Human Activity Recognition (HAR), most existing studies rely on wearable sensors for the detection of simple activities, which are inadequate to support the detection of complex activities. Therefore, further sources (e.g., environmental sensors and home appliance sensors) are incorporated to help the system's accurate detection of complex activity. In our previous work [12], we designed a preliminary reminder care system based on the detection of complex activities. We conduct recommendations via three main steps:

- Elementary activity recognition: In this approach, the common configuration of DeepConvLSTM was used as the classifier to detect elementary activities. The DeepConvLSTM was configured to four convolutional layers with feature maps and two LSTM layers with 128 cells. This stage was tested on two public datasets PAMAP2 dataset [33] and PUCK dataset [34]. The result shows that DeepConvLSTM achieved a promising accuracy of 77.2%.
- Ontology for complex activity recognition: After achieving the detection of elementary activities, we built an OWL (OntologyWeb Language3) model, which includes the artefacts, locations, environment, and activities required to define things involved in the interaction. From the Aris scenario, preparing a cup of tea could involve changes in the motion sensor (local environmental sensor), status of the kettle in triggering the usages, and time period for this activity, which would rarely be in the early morning before sunrise. From the example, we can extract numbers of context: First, from the motion sensor referring to Aris' place (the kitchen); Secondly, the item context where the kettle has been used; and finally, the time context of when this activity took place.
- Rule-based orchestration: This step utilizes the output from the two previous steps for the detection of complex activities. A set of rules produced based on the previous ontological models are implemented. Following the Aris tea preparation illustration, we can create an ontological rule in a descriptive language as:

$$
\begin{aligned}
&PreparingTea\\
&\quad\sqsubseteq\ Cooking\\
&\qquad\sqcap\exists\ involving\ \text{Artefact.Kettle}\\
&\qquad\sqcap\exists\ user\_is\_is\ \text{Area.Kitchen}\\
&\qquad\sqcap\exists\ user\_is\_conducting\ \text{Activity.Standing}\\
&\qquad\sqcap\neg\exists\ has\ \text{Time\_constraint.EarlyMorning}
\end{aligned}
$$

### 3.2. Prompt Detection

At this stage, the data collected from the previous stage is used to determine the prompt of an activity. The prompt is considered in two main situations: when the user appears to be stuck within an activity for notable period without taking an action, and when the user uses a wrong item that does not belong to this activity. In the previous stage, the developed complex activity detection module is capable of complex activity detection and learning of different activity patterns. The extracted features of each activity are used to build the prompt detection system.

As mentioned earlier, Alzheimer patients in the mild stage may not be able to complete their activities due to forgetfulness. For example, if Aris forgets to turn off the stove after making her tea, the system can detect that the user needs a prompt and present an immediate recommendation to turn off the stove. Various learning models are applied in the determination of when a user needs a prompt during a monitored activity. Das et al. [34] tested several classification algorithms (Support Vector Machines (SVM) [35], Decision Tree [36] and Boosting [37]) on the PUCK dataset.

In particular, Boosting applies a classification algorithm to re-weight the training data versions sequentially and then extract a weighted majority vote of the previous sequentially classifiers. It generally outperforms the other two methods. For our experiments, only one dataset provides the labels where the user needs a prompt or not by adding a class from 0 and 1. However, for the other two datasets, we create the points that define when the user need prompt as we explain in detail in Section 4.

### 3.3. Conducting Recommendations

Having determined that a user's activity requires a prompt, the system at this stage then decides which item can be suitably recommended based on the user situation. One of the main challenges is handling each activity differently. The system must always consider what correct item is to be recommended even if it is the same activity by considering the

user situation. For this reason, each activity is treated differently as a session during the training where it helps the system to learn different pattern of each activity. This stage represents our main contributions in this paper.

### 3.3.1. Problem Definition

When a complex activity that needs a prompt $v$ where $v_i \in V = \{v_1, v_2, .....v_m\}$ is received by the agent $G$ at time $t$, our algorithm extracts the context $x$ and nominates an appropriate item $a$ from a set of actions $A$ for the current activity. Notice that our system recommends only one item at each time that needs a prompt. Then, the agent receives feedback as reward $r$ for the recommended item. Finally, the system is being updated based on the received reward, which is called minor updated $Miu$, and the major update is after a certain time of period $Mju$. Table 1 summarises the notations used throughout this paper.

**Table 1.** System notations.

| Notation | Explanation |
|:---:|:---|
| $G$ | Agent |
| $x, X$ | Context, set of context |
| $a$ | Action(item) |
| $r$ | Reward |
| $A$ | Set of items |
| $M$ | Memory |
| $S$ | State |
| $Miu$ | Minor update |
| $Mju$ | Major update |
| $T_r$ | Reward Delay Period |
| $\prod$ | Policies |
| $SV$ | State value of each sensor |

### 3.3.2. Method

The problem is formulated based on a contextual bandit approach to tackle the dynamicity of human activity patterns and to recommend the correct item without having to wait for the user's feedback. Contextual bandit provides a learning model based on context. Three kinds of context are extracted at this stage:

- **Past activities context (PAC)**: Note that each activity is desired to have a different pattern; thus, for each activity, the system extracts the path/sequences of items used from the past records (recorded in the log file) as a type of context. The observed paths of each activity are then stored in a memory based on which the agent can decide an item to be recommended at a specific situation.
- **Current activity Context (CAC)**: The contexts on the current states are extracted from the received data obtained from the previous two stages. For example, when the system receives that the user needs a prompt for preparing coffee, the context of the current activity (locations, previous items, user position and time) will be extracted.
- **Item context (IC)**: This essentially concerns information about items, such as determining to which activity an item belongs, how long such an item can be in use, and how many times such items are needed by the user for the current activity. For example, a coffee machine as an item can be used for the activity of 'preparing coffee', where it can be used for around 2 min each time.

The contextual features of each session of the activities are received by the agent as input (Algorithm 1). The CB combined three main components: an environment, which represents the context of the user's activity $x \in X$, an agent G, which chooses an action $a \in A$, which is represented as an item in our system (notice that the common name in CB is Learner but we call it an agent in our case) based on the received context and a

reward $r \in [0,1]$, which the agent aims to maximize by recommending the correct action at each round t= 1, 2, ...,T. We calculate the expected reward after each recommendations as follows:

$$r(a) = \frac{\sum_{t=0}^{T_r} SV_{a,t}}{\sum_{a=0}^{A} \sum_{t=0}^{T_r} SV_{a,t}} \tag{1}$$

where $T_r$ is the Reward Delay Period (this is explained below) and $SV$ depicts the state value of each sensor at each time step $t$. The reward function for automatically obtaining the feedback without waiting for the user is formulated. As shown in Figure 4, the calculated reward gives a non-correct recommended item (the coffee machine). Consequently, the system considers it for updating.

Two types of updates are carried out to keep the system in constant interactive recommendation, minor update and major update. In minor update *Miu*, the system is updated after the agent receives the reward where the system compares the recommended item with the item already used by the user instead of it. Such item is then updated with the newly used item to keep tracking the remaining items of the activity.

---

**Algorithm 1** Our procedure to recommended a correct item for user's activity. It takes context $x$ as input, and returns a recommended item as output $a$.

---

  1: Initialize the capacity of storage memory $M$
  2: Initialize a timer$= 24h$
  3: **for** session$v_t \in V$ **do**
  4:     Observe state $s_t$
  5:     Extract $x_t$, where $x_t = PAC, CAC, IC$
  6:     Execute action following set of policies $\prod$
  7:     waiting for $T_r$
  8:     Compute $r(a)$ according Equation (1)
  9:     Minimize $R$ according Equation (2)
10:     update $Miu$
11:     Put $x_t, r_t, a$ into $M$
12:     Update $Mju$
13: **end for**
14: **return**

---

Major updates are considered after certain period before the end of the day where the system is updated using the memory of historical data as obtained from the agent in the last 24 h. The major update *Mju* helps the system to tackle any dynamicity of the user pattern during the day as we mentioned before in the Aris scenario that her pattern could be changed even if she has still doing the same daily activities.

Most traditional recommender systems focus on 'click' or 'not click' as feedback to promptly evaluate the reward function and to update the system. In contrast, our system having recommended an item, waits for sufficient time to determine if the recommended item is used or not by checking its status (on/off or moved/not moved). This status is then used to update the system accordingly. Furthermore, if the system recommends a coffee machine to Aris (see Figure 4)) when she is preparing a cup of coffee, whereas she wants to use it later and not on the immediate.

The recommended item may not be seen as incorrect due to the false negative feedback at this time. The recommendation though not needed at this time can be used at another time. To facilitate the above, we introduce a Reward Delay Period $T_r$, which accounts for the different paces of users in carrying out activities, and we consider $T_r$ a hyperparameter (to be detailed Section 5).

The agent G can choose from a set of policies $\prod \subseteq \{x \rightarrow A\}$ to map each context for a suitable item by employing two streaming models: Linear regression and stochastic gradient distance (to be detailed in Section 5). The goal of using different polices is to minimize the regret $R$ between the expected reward of the best action $a^*$ and the expected reward of selected action $a$. The regret is calculated by using the following equation:

$$R_t(T) = E\left[\sum_{t=1}^{T} r_t, a_t^*\right] - E\left[\sum_{t=1}^{T} r_t, a_t\right] \tag{2}$$

Here, we adapt three categories of policies as the following:

- Randomized (AdaptiveGreedy), which focuses on taking the action that has the highest reward.
- Active choices (AdaptiveGreedy), which is the same for AdaptiveGreedy but with active parameter ! = None, which means actions will not be taken randomly.
- Upper confidence bound (LinUCB), which stores a square matrix, which has dimension equal to total numbers of features for the fitted model. Details about the parameters for each policy of two streaming models: Linear regression and stochastic gradient distance will be detailed in Section 5.

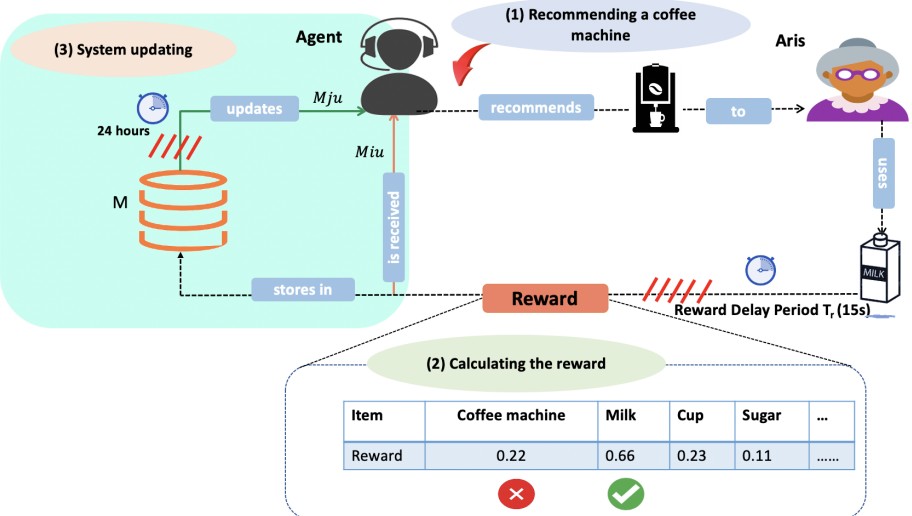

**Figure 4.** (1) The agent recommends a coffee machine to Aris, whereas she uses milk instead; then the system waits for 15 s; (2) The feedback is received by the system as a reward, and it is calculated accordingly as the coffee machine is the wrong item; and (3) two kinds of updating for the system: a minor update is after receiving the reward and a major update after certain period around 24 h.

## 4. Dataset

We have to mention that our evaluation focused on the third stage of our system, which is the conduction recommendation. This was evaluated on three public datastets: PUCK [34], ARAS [38] and ADL [39].

### 4.1. PUCK Dataset

The PUCK dataset is a public dataset published in 2011. The PUCK dataset collected from a Kyoto smart home testbed located in Washington State University in two-story apartments with one living room, one dining area and one kitchen on the first floor as well as one bathroom and three bedrooms on the second floor.

It combines three types of sensory data: (1) environmental sensors, including motion sensors on ceilings, door, sensors on room entrances, kitchen cabinet doors, microwave, and refrigerator doors, temperature sensors in rooms, power meter, burner sensor, water usage sensors, and telephone usage sensors, (2) items sensors for usage monitoring, and (3) two wearable sensors. Eight complex activities are defined: Sweep and Dust, DVD Selection and Operation, Prepare Meal, Fill Medication Dispenser, Water Plants, Outfit Selection, Write Birthday Card, and Converse on Phone. Activities are divided into ordered steps, which can help detect whether the activity is completed correctly.

Features Engineering

The PUCK dataset has four fields (date, time, sensor ID and sensor value). To adapt the PUCK dataset for our system, the following steps were taken to process the PUCK dataset by extracting the required features:

1.  Combining the environmental data sensors (motion, items, power meter, burner, water usage, door etc.) with the wearable sensors for each participant by matching the time step among them.
2.  Labelling complex activities for the whole dataset.
3.  Extracting the start and the end of each activity as a session to define when the user needs a prompt.
4.  Selecting only the common sensors among all participants where the total measurement counts and participants each greater than 25th percentiles.
5.  Dividing the sensors into four groups to be processed: movement sensors, motion sensors, count sensors and continuous values sensors and process each group as follows:

    (a)  In movement sensors group, each measurement includes six values (X, Y, Z, Yaw, Roll and Pitch). We extracted the following features: Mean (X, Y, Z, YY, RR and PP), STD (X, Y, Z, YY, RR and PP) Correlations (X//Y//Z) and (Yaw//Roll//Pitch), which leads to 36 features in total.
    (b)  For motions sensors group, if at least one trigger in a group is counted as trigger for the group, count and then compute the fraction counts across the groups. Based on the PUCK dataset, we have 11 groups (features) altogether.
    (c)  Count sensors, which have on, off measurements, such as (door, item, shake and medicine container sensor), we count and compute the fraction counts of each session (20 features).
    (d)  For the last group, we calculate the average for continuous value sensors, such as electricity and temperature (three features).

6.  After extracting the all features (70 features), we apply the previous groups process for all the participant sessions.

Two methods are taken to overcome the item usage imbalance problem (i.e., only a small number of items are frequently used): first, the outliers from the items are dropped. This method is simple but very effective in improving the performance and secondly, the sampling order random points in activity sessions to increase the prompt points, although this does not help balance the item usages, such as dropping the outliers.

*4.2. ARAS Dataset*

The ARAS dataset consists of 2 h of two residents where they perform 27 daily living activities: Going Out, Preparing Breakfast, Having Breakfast, Preparing Lunch, Having Lunch, Preparing Dinner, Having Dinner, Washing Dishes, Having a Snack, Studying, Having a Shower, Sleeping, Watching TV, Toileting, Napping, Brushing Teeth, Using the Internet, Laundry, Shaving, Cleaning, Talking on the Phone, Listening to Music, Having Conversation, Reading a Book, Having a Guest, Changing Clothes and Others.

The activity sensory data is collected from 20 binary sensors, including force sensitive resistors (FSR), pressure mats, contact sensors, proximity sensors, sonar distance sensors, photocells, temperature sensors and infrared (IR) receivers. Due to the differences between house A and house B, every house has a different topology of the Wireless Ambient Sensor Networks (WASN) where house A has two of the Personal Area Networks while house B has only one. The sensory data is collected for full month for each house with the time stamp of one second.

Features Engineering

The ARAS dataset consists of 22 columns where the first 20 columns represent sensors and the last columns represent the activities labels of each resident. We added a new

column for the time to facilitate the feature engineering. The ARAS dataset does not require complex feature engineering process because it has binary values for the whole sensors. For both houses, the activities that interact with items sensors are used for the experiments and others are removed. Unlike the PUCK dataset, the ARAS dataset does not contain the prompt points. Based on that, we added random prompt points for each activity (session), and we consider them to be before the session is ended around the reward delay period $T_r$.

### 4.3. ADL Normal Dataset

The ADL Normal dataset represents a public dataset published in 2010. It was collected from a Kyoto smart apartment testbed in Washington State University. The data contains 20 participants performing five complex activities that are defined as: making a phone call, washing hands, cooking a meal and eating, taking medicine and cleaning. It is collected per second and annotated using the activity number and number of each participant. The ADL Normal combines only motion sensors, item sensors, burner sensors, phone usage and water sensors.

### Features Engineering

The same features engineering of the PUCK dataset is applied except that of the wearable sensors processing. We add random prompt points for each activity but considering that these points should be before the session is ended, we follow the same process as for the ARAS dataset.

## 5. Evaluation

First, the evaluation of the effectiveness of the CB approach in recommending the correct item to a user in case the user's current activity needs a prompt is carried out. All the extracted features are utilized by the system as contexts to make a recommendation of the correct item. One publicly available CB python package is selected for our experiments. The package offers two types of models: full batch and streaming models. Due to sample limitations of the datasets, the streaming models, namely SGDClassifier (SGD) and LinearRegression (OLS), are focused on.

Both models are sensitive to hyper-parameters, such as beta_prior or smoothing. Nevertheless, SGDClassifier offers stochastic matrices, while LinearRegression (OLS) has matrices, which are closed to the solution, and it updates them incrementally. Details about the parameters are given in Table 2. Figure 5 shows a set of policies used for each model. For the PUCK Figure 5a and the ARAS (house A) (Figure 5b) datasets, the SGD model—particularly the Softmax Explorer policy—is more robust, and it provides a better cumulative mean reward of item recommendations.

On the other hand ARAS (house B) Figure 5c and ADL Figure 5d are more likely to provide good results by LinUCB policy of OSL model. This plot confirms that both models lead to a promising results based on each dataset from several policies being used. Table 3 shows the cumulative mean reward of our system by considering the three datasets.

The Reward Delay Period $T_r$, as earlier explained, helps in the determination of the suitable time for an agent to receive the reward as a feedback of the recommended item. Tuning $T_r$ is important as decreasing it could indicate that the recommended item is not used, while increasing $T_r$ could confuse the agent—specifically when the user starts to use other items before receiving the feedback about the recommended one. The results in Figures 6–8 show how $T_r$ can affect the performance.

The reward delay period with 5 s (see Figure 6) provides higher performance with the three datasets compared with the reward delay period of 10 and 15 s as shown in Figures 7 and 8, respectively. Policies are affected by the reward delay period; for instance, softmax explorer of the SGD model has a good performance with 5 s among the datasets but this performance starts to be reduced when the reward delay period is increased to 10 and 15 s. Table 4 summarises all cumulative mean rewards of our system among three datasets using different reward delay periods.

There are two reasons that make the reward delay period of our system vary from dataset to another or from user to user in real time system. The first reason is that each activity has specific items and each of these items existing in different place. For example, preparing a cup of coffee activity includes the following items: a cup, sugar, milk and a coffee machine. Thus, if the agent recommends milk and the fridge is too far from the user, it will take some time to receive the correct response, while if the agent recommends a cup and the user is standing near the cupboard, it will take less time to receive the response.

The other reason is that users have different behaviours in their response; some of them respond immediately after they receive the recommended item, and others may take a little bit longer. However, our system is targeted to Alzheimer's patient where it should expect a reward delay period with a long time compared with a healthy person. Here, we treat $T_r$ as a hyperparameter that can be adjusted based on each item; we will leave this to our future work. In addition, it is observed that the system achieved the desired result of not requiring any feedback from the user to receive the reward. Consequently, it is calculated automatically after the Reward Delay Period. This feature is focused on because our system deals Alzheimer's patients who experience difficulty in holding a smartphone and confirm their response for recommendations.

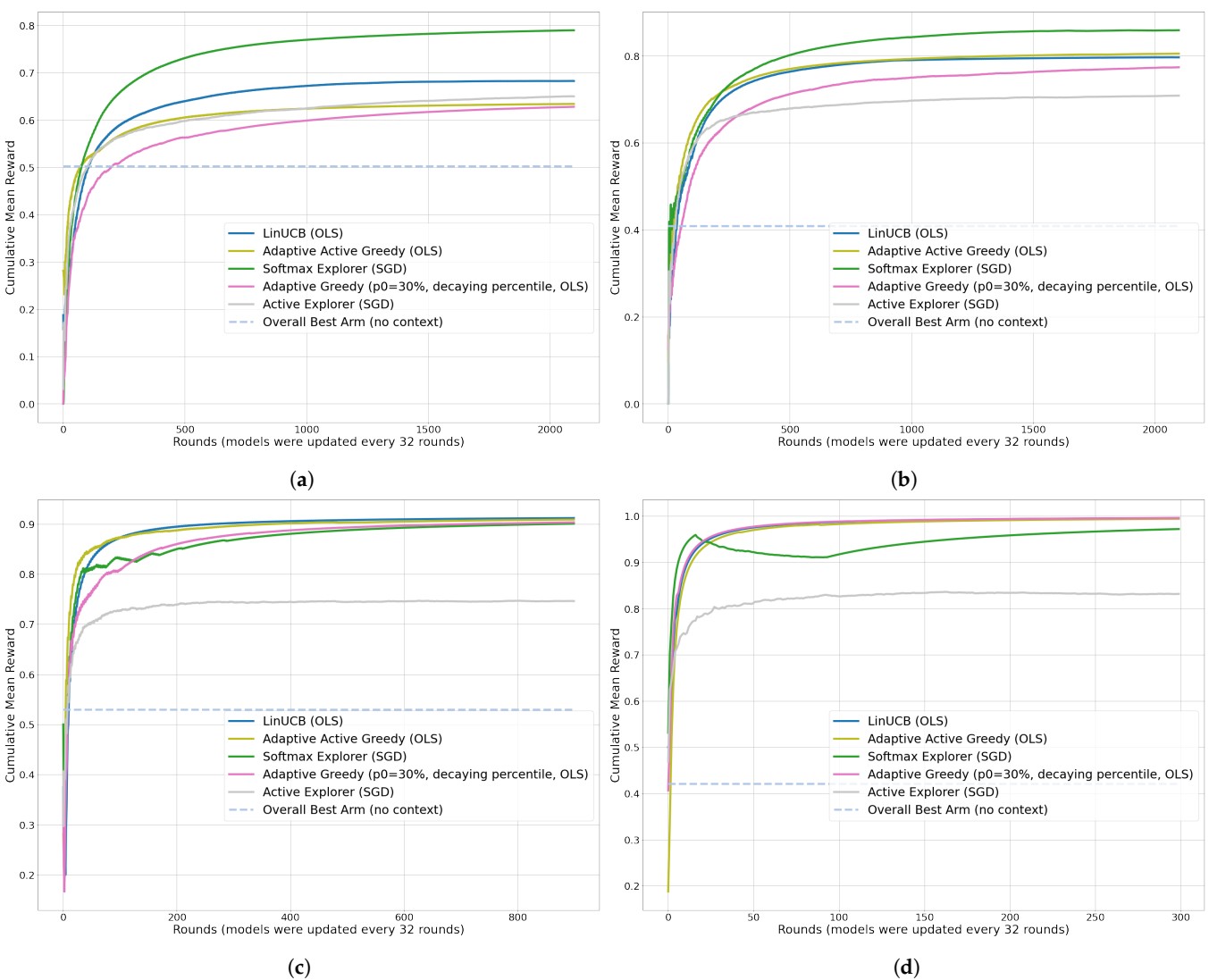

**Figure 5.** The comparison of models for each dataset. (**a**) PUCK; (**b**) ARAS (house A); (**c**) ARAS (house B); (**d**) ADL.

**Table 2.** Tuning hyperparameters for the OSL model policies.

| Policy | Note | Hyberparameters | | | | | | |
|---|---|---|---|---|---|---|---|---|
| | | Beta_PRIOR | Alpha | Smoothing | Decay | Refit_BUFFER | Active_CHOICE | Decay_TYPE |
| LinUCB [30] | LinUCB policy stores a square matrix , which has dimension equal to total numbers of features for the fitted model. | None | 0.1 | - | - | - | - | - |
| AdaptiveGreedy [40] | It focuses on taking the action that has the highest reward. | None | - | (1,2) | 0.9997 | - | - | percentile |
| AdaptiveGreedy(Active) | It is the same for AdaptiveGreedy but with different hyberparameters | ((3./nchoices, 4), 2) | | None | 0.9997 | - | weighted | percentile |
| SoftmaxExplorer [40] | It depends on softmax function to select the action | None | - | (1,2) | - | 50 | - | - |
| ActiveExplorer [40] | It depends on an active learning heuristic for taking the action | ((3./nchoices, 4), 2) | - | None | - | 50 | - | - |

**Table 3.** The cumulative mean reward of our system among three datasets.

| Dataset | Policies | | | | |
|---|---|---|---|---|---|
| | LinUCB (OSL) | Adaptive Active Greedy (OLS) | Adaptive Greedy (OSL) | Softmax Explorer (SGD) | Active Explorer (SGD) |
| PUCK | 0.68 | 0.64 | 0.63 | **0.79** | 0.65 |
| ARAS House (A) | 0.80 | 0.81 | 0.77 | **0.85** | 0.69 |
| ARAS House (B) | **0.92** | 0.91 | 0.91 | 0.90 | 0.75 |
| ADL | **0.99** | **0.99** | **0.99** | 0.97 | 0.83 |

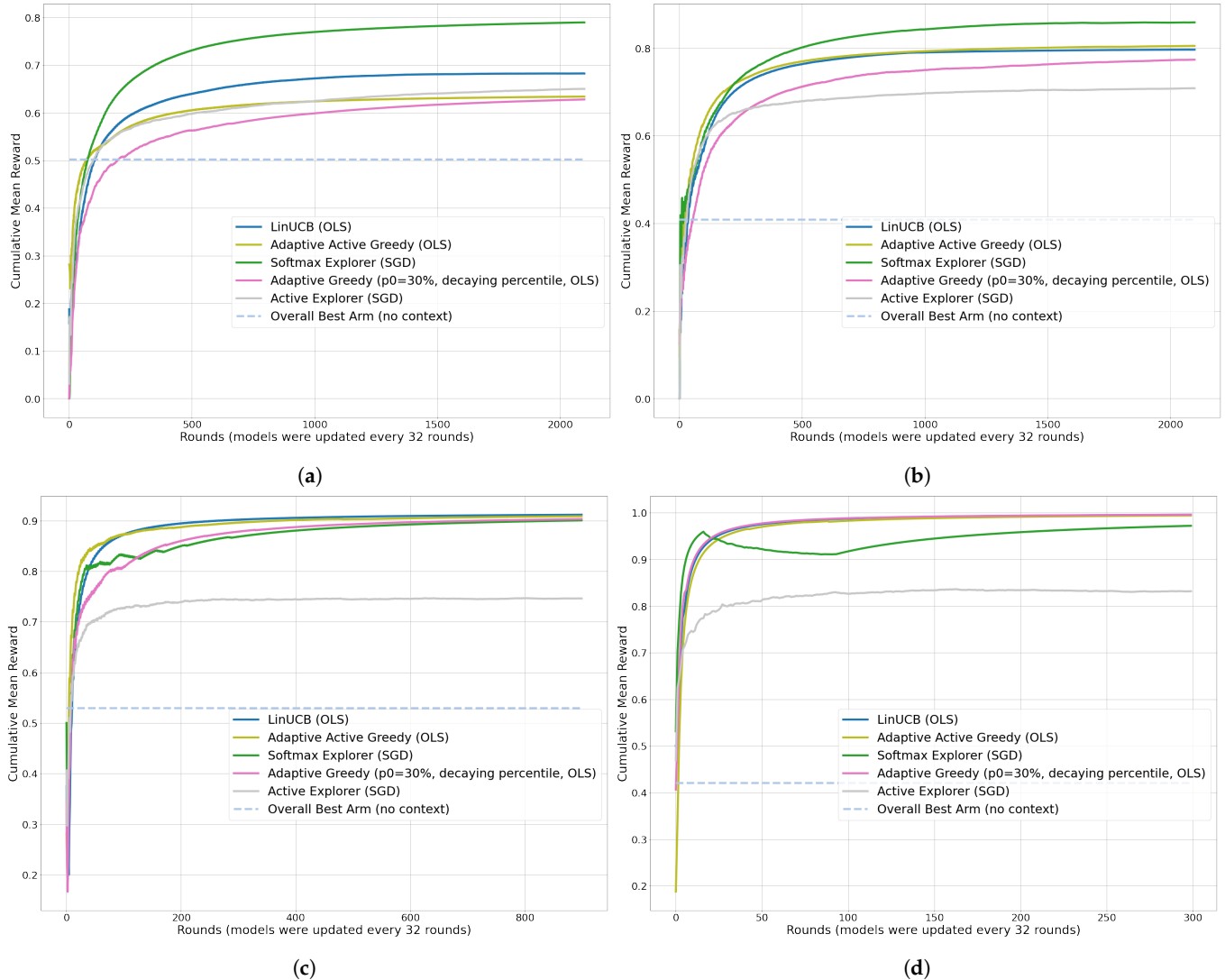

**Figure 6.** The Reward Delay Periods $T_r = 5$ s. (**a**) PUCK; (**b**) ARAS (house A); (**c**) ARAS (house B); (**d**) ADL.

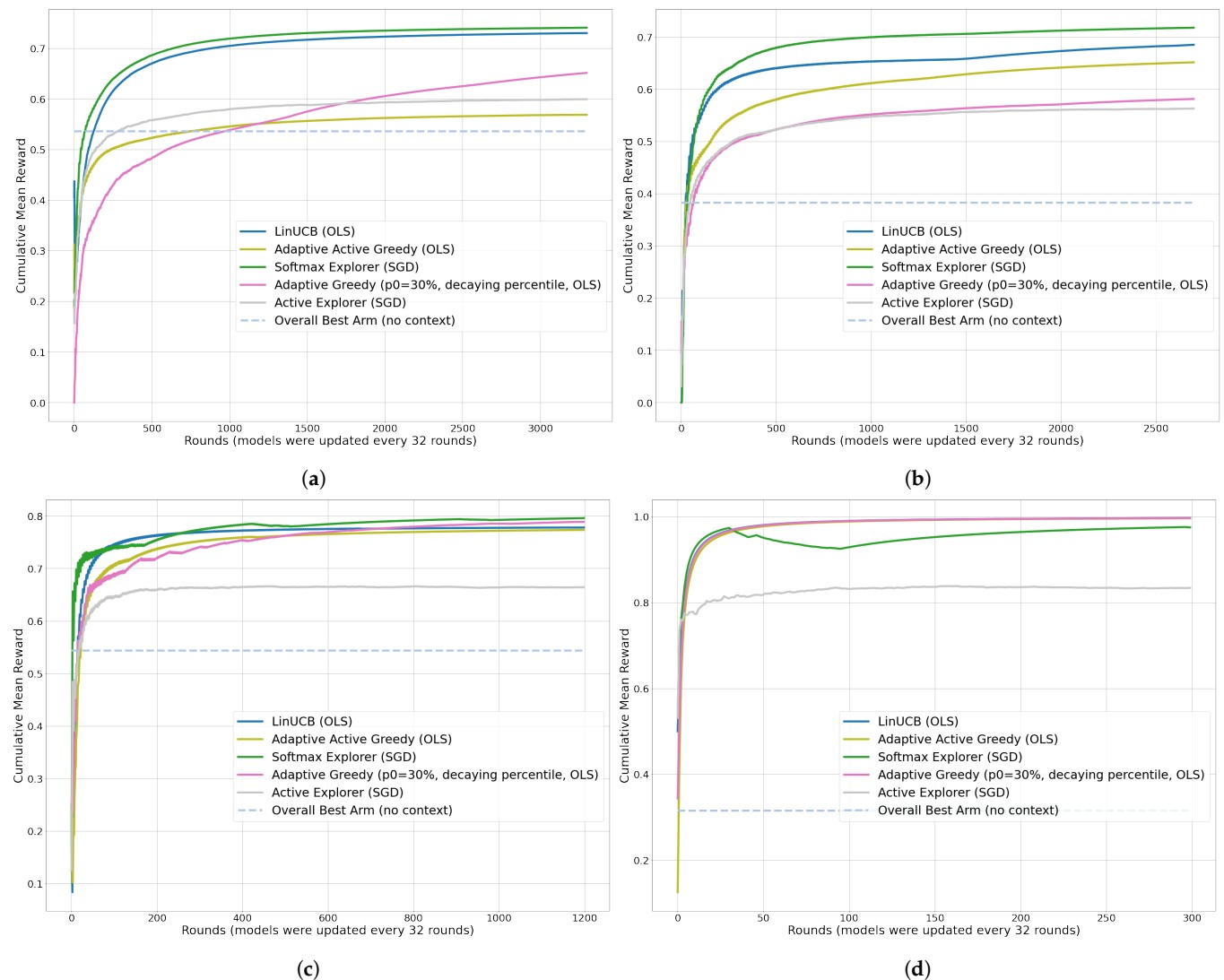

**Figure 7.** The Reward Delay Periods $T_r$ = 10 s. (**a**) PUCK; (**b**) ARAS(house A); (**c**) ARAS (house B); (**d**) ADL.

**Table 4.** The cumulative mean reward of our system among three datasets using different reward delay periods.

| Dataset | The Reward Delay Period | Policies | | | | |
|---|---|---|---|---|---|---|
| | | LinUCB (OSL) | Adaptive Active Greedy (OLS) | Adaptive Greedy (OSL) | Softmax Explorer (SGD) | Active Explorer (SGD) |
| PUCK | 5 s | 0.68 | 0.64 | 0.63 | **0.79** | 0.65 |
| | 10 s | 0.74 | 0.55 | 0.65 | **0.75** | 0.60 |
| | 15 s | 0.62 | 0.52 | 0.48 | **0.70** | 0.58 |
| ARA s Hou se (A) | 5 s | 0.80 | 0.81 | 0.77 | **0.85** | 0.69 |
| | 10 s | 0.68 | 0.65 | 0.58 | **0.72** | 0.56 |
| | 15 s | 0.68 | **0.72** | 0.68 | 0.60 | 0.49 |
| ARA s Hou se (B) | 5 s | **0.92** | 0.91 | 0.91 | 0.90 | 0.75 |
| | 10 s | 0.76 | 0.76 | 0.78 | **0.79** | 0.66 |
| | 15 s | 0.71 | 0.65 | 0.76 | **0.74** | 0.61 |
| ADL | 5 s | **0.99** | **0.99** | **0.99** | 0.97 | 0.83 |
| | 10 s | **0.99** | **0.99** | **0.99** | 0.98 | 0.83 |
| | 15 s | **0.95** | **0.95** | **0.95** | 0.90 | 0.78 |

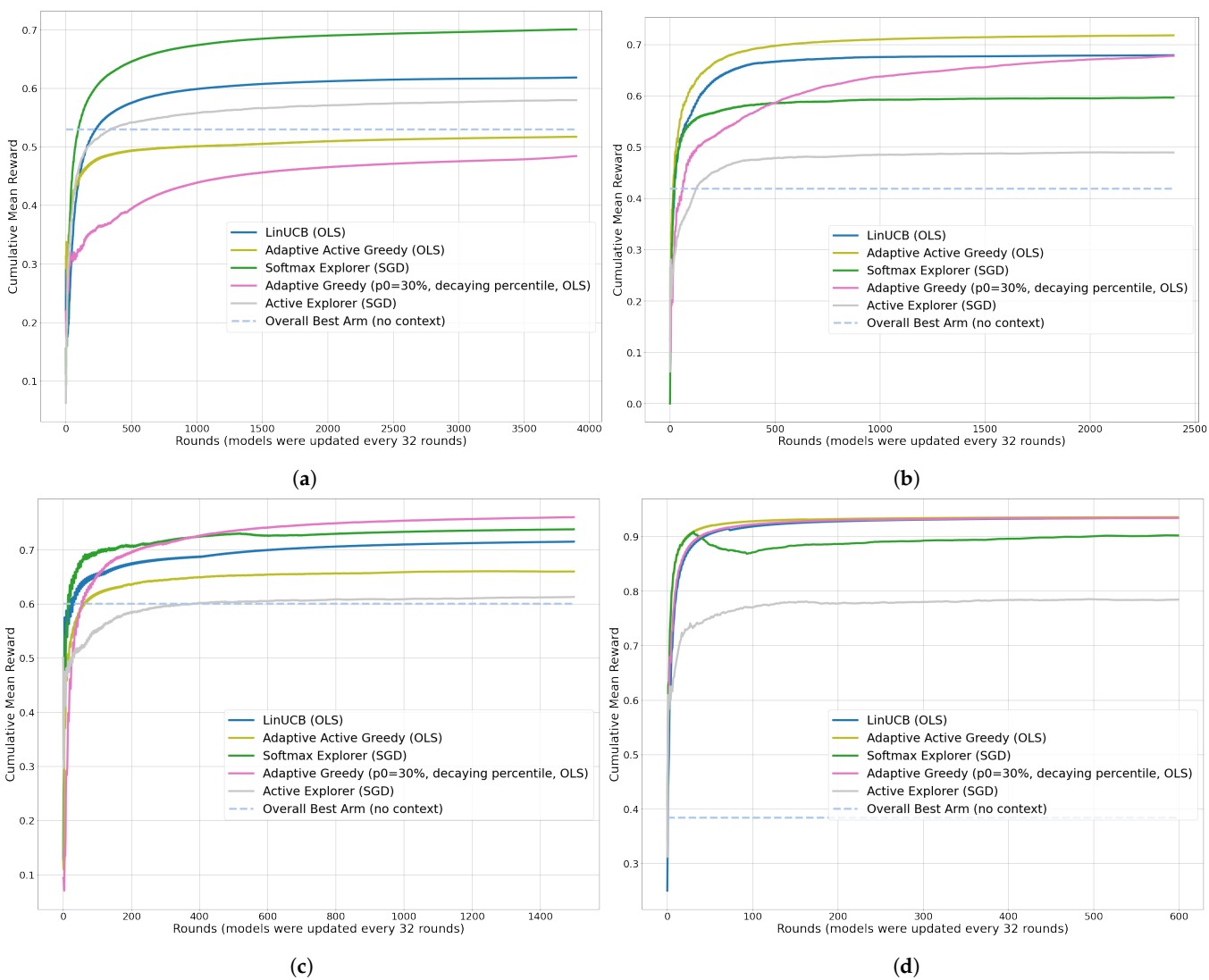

**Figure 8.** The Reward Delay Periods $T_r$ = 15 s. (**a**) PUCK; (**b**) ARAS (house A); (**c**) ARAS (house B); (**d**) ADL.

## 6. Scope of Improvements Directions for RCS

Despite all the advantages of the proposed system, there are still aspects that need to be considered in the future. Some of them are discussed below.

- **The RCS with real life**. As mentioned, our system was only tested on public datasets. Dealing with real time data, our system should be capable of synchronization among the three stages starting from the complex activity detection until the user receives an item recommendation. We need to build our model for prompt detection that can exactly define when the user needs a recommendation. Failure in this task makes the system construct not beneficial recommendations that could affect the quality of the system.

- **RSC testbed**. Building a testbed helped to evaluate our system in the real life. The main issue with public datasets is missing required features. For example, the time period of each activity as some activities rarely happen at night time, such as Aris preparing a cup of coffee at midnight. Thus, if the system was feeding with the time period of each activity, it will be expected to recommend going back to bed for Aris and to mention the time to remind her.

- **Trust-aware of the recommendations**. Our system deals with sensitive and critical data about the patient, a lack of integrity could harm the user's life by suggesting

incorrect items, such a recommending a medicine when the user has already taken it. To ensure the safety of the recommendations, the data that feeds our system needs to be protected.

The blockchain is planned as a potential step forward to address the integrity challenge. Our previous work [41] introduced a conceptual framework for data integrity protection.

- **Unexpected action**. In some statuses, our system could face an issue when the user uses two items at the same time, and there is only a short time period between them. This case could make the agent receive wrong feedback about the recommended item, which could affect the system update. For example, if the agent recommends turning the coffee machine on, whereas the user brings the milk at the same moment and then accepts the recommendation. After calculating the reward, it seems that is the milk is the correct item not the coffee machine.

- **Easy to handle**. As we mentioned before, we targeted Alzheimer's patient in the mild stage; therefore, our system should consider that elderly people cannot hold a phone to receive the recommendations. Consequently, designing a system that acts as caregiver for the patients is important to meet the user's expectations.

## 7. Conclusions

In this work, the feasibility of building a reminder recommendation system was explored. The recommendation system was adapted for Alzheimer's patients for when they need a reminder. We took advantage of the contextual bandit approach to formulate our problem and tackled two main issues: the dynamicity of human activity patterns and recommending the correct item without needing explicit user feedback. Experiments demonstrated the effectiveness of our recommender system. Some limitations observed in our evaluation of the system include that our experiments are still not sufficiently comprehensive because the datasets that we used did not meet our system's requirements, such as time labels, which are an important and critical type of context.

Only the PUCK dataset, which considers the wearable sensors as a source to detect complex activity, was analysed; however, the other two datasets included items and environment sensors. The number of samples and complex activities in each dataset were also considered as limitations that affected our experimental results. In the future, we will create our own test-bed to collect inclusive and adequate data for complex experiments and test our framework in real-life scenarios.

**Author Contributions:** Methodology, M.A. and L.Y.; Supervision, C.H., X.W. and S.S.K. All authors have read and agreed to the published version of the manuscript.

**Funding:** This research received no external funding.

**Data Availability Statement:** Not Applicable, the study does not report any data.

**Conflicts of Interest:** The authors declare no conflict of interest.

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
