# Peer review of "Context-Induced Activity Monitoring for On-Demand Things-of-Interest Recommendation in an Ambient Intelligent Environment"

_futureinternet, doi:10.3390/fi13120305_

Round 1
Reviewer 1 Report
This is the review of the manuscript entitled "Context-induced Activity Monitoring for On-demand Things-of-Interest Recommendation in an Ambient Intelligent Environment".
The subject may attract interest to the readers. The authors present an interesting topic, being in line with the mission of the Future Internet Journal. In general, this manuscript is well organized and written, with a comprehensive literature review, detailing the framework approach of the study, clearly stated methodology and nicely presented findings. The manuscript provides sufficient background information regarding the topic proposed.
I have some suggestions:
- The abstract section is missing some quantitative main research findings.
- The authors can highlight the usefulness of the study in its practical applicability.
- Providing information regarding how the accuracy of the results was verified.
Author Response
Dear Reviewer,
Thank you for your time and efforts in reviewing this manuscript. The comments are very
constructive and valuable, and we have considered these comments seriously in revising the
manuscript. Here, we respond to the review comments one by one as follows:
comments
• The abstract section is missing some quantitative main research findings.
Response: Thanks for your valuable suggestion, in the abstract, we added the evaluation metric that we
used in our experiments.
• The authors can highlight the usefulness of the study in its practical applicability.
Response: we highlighted it in the introduction section by illustration Aris scenario [from line #39 to #47]
• Providing information regarding how the accuracy of the results was verified.
Response: we verified it by comparing the cumulative mean reward using different policies
on the three public dataset. Also, we evaluated the system based on the reward delay period and we explained
the reasons of the differences [from line438 to 454]
Reviewer 2 Report
The goal of this paper, as exposed by the authors, is to present a reminder care system to help Alzheimer patients live safely and independently in their homes.
Section 1 and 2 contain the introduction and the review of state of the art. It was successfully presented the current field of research, improvements and current solutions. It can still be approached the IoT technological limitations and research directions. Related work section contain actual works in the field, and should be improved with a comparative study between them.
What are the real-time aspects that characterize the RCS system proposed in the paper? What are the security aspects of the real-world IoT based smart home applications? Section 5 must be completed in this respect.
The contributions, practical results, as well as the way in which they were obtained, are clearly presented. Which are the future work in the context of healthcare IoT, human activity patterns, and complex activity recognition scientific research? How can the problems given by the number of samples and complex activities be solved in the future?
Author Response
Dear Reviewer,
Thank you for your time and efforts in reviewing this manuscript. The comments are very
constructive and valuable, and we have considered these comments seriously in revising the
manuscript. Here, we respond to the review comments one by one as follows:
comments
• Related work section contain actual works in the field, and should be improved with a
comparative study between them.
Response: Thanks for pointing this out, in related work section#2, we explained the issue of each current
approach and how the proposed approach can address these issues (from line #129 to line 131), (from
line#140 to line#142) and (from line#153 to line#157)
• What are the real-time aspects that characterize the RCS system proposed in the paper?
What are the security aspects of the real-world IoT based smart home applications? Section 5
must be completed in this respect.
Response: the real-time aspects of our system that it can deal with the patients as caregiver by
addressing two main issues that other approaches are faced (in section#1 from line #77 to line 79).
In this work we don’t consider the security issue, however, we highlighted it in section#6 as a future work and
we citied one of our work that provides a conceptual framework to address one of the security issues (from line
#471 to line 477)
• Which are the future work in the context of healthcare IoT, human activity patterns, and
complex activity recognition scientific research? How can the problems given by the number
of samples and complex activities be solved in the future?
Response: we mentioned our previous work which provides our method for detecting the complex activities
in subsection (3.1. Complex activity detection) line#201 also, we mentioned our testbed in section#7 (from line
#502 to line#504)